# $\mu$-Parameterization for Mixture of Experts

## Abstract

Recent years have seen a growing interest and adoption of LLMs, with Mixture-of-Experts (MoE) emerging as a leading architecture in extremely large models. Currently, the largest open-source models reach over 1T parameters. At such scales, hyperparameter tuning becomes prohibitively expensive. Precisely for this reason, the $\mu$Transfer is becoming a key technique. It allows for seamless transfer of optimal hyperparameters across model scales, resulting in a huge reduction in tuning costs. However, existing work has primarily focused on dense LLMs, leaving MoE architectures unexplored. In this work, we derive a $\mu$-Parameterization for MoE, providing theoretical guarantees for feature learning across model widths. Our experiments demonstrate that the optimal learning rate reliably transfers across model sizes, establishing a foundation for efficient hyperparameter tuning in large-scale MoE models.

## 1 Introduction

Scaling large language models (LLMs) has been a central driver of recent progress in AI. Scaling laws show that larger models trained for longer consistently achieve better performance. This naturally places us in a regime where computational resources are perpetually the limiting factor. In such settings, the efficiency of hyperparameter tuning is as important as the efficiency of the final training run, since both draw from the same compute budget.

The $\mu$-Parameterization ($\mu$P) (Yang, 2021) provides a principled framework for stable and predictable training dynamics across model widths. By reparameterizing models to preserve feature learning in the infinite-width limit, $\mu$P ensures that optimal hyperparameters (e.g., learning rates, initialization scales) remain invariant to model width. This allows hyperparameter searches to be conducted on small models, saving most of the computational resources for the final training run.

In parallel, Mixture-of-Experts (MoE) has become one of the most widely adopted architectural innovations for improving the model efficiency (DeepSeek-AI et al., 2025; OpenAI et al., 2025; Team et al., 2025). MoE decouples parameter count from computational cost through conditional computation, replacing each dense feedforward block with a router and a set of expert networks. Despite its popularity and effectiveness, MoE has not yet been parameterized for feature learning in the infinite-width limit, leaving open the question of whether $\mu$P can be extended to sparse architectures of this kind.

In this work, we extend $\mu$P to MoE architectures, providing both theoretical grounding and empirical validation. Our main contributions are:

- **Parameterization with learning rate transfer in MoE models.** We find a parameterization for MoE models that transfers the optimal learning rate across model widths. We empirically verify this transferability.

- **Theoretical grounding of $\mu$P for MoE.** Building on Yang et al. (2022), we provide a theoretically principled derivation of a parameterization that ensures feature learning is preserved across all weights within MoE models.

- **Limits of transferability.** We show that while learning rate transfer holds across widths, it breaks down when varying the granularity of MoE, identifying boundaries for hyperparameter transfer in MoE.

| Component | Init. Var. | Multiplier | LR (Adam) |
|---|:---:|:---:|:---:|
| Embedding | 1.0 | 1.0 | 1.0 |
| Unembedding | 1.0 | $1/d_{\text{input}}$ | 1.0 |
| Attention (Q, K, V, O) | $1/d_{\text{input}}$ | 1.0 | $1/d_{\text{input}}$ |
| Feed-forward (dense) | $1/d_{\text{input}}$ | 1.0 | $1/d_{\text{input}}$ |
| Experts (MoE) | $1/d_{\text{input}}$ | 1.0 | $1/d_{\text{input}} \mid 1/d_{\text{input}}$ |
| Router (MoE) | $1/d_{\text{input}} \mid 1.0$ | $1.0 \mid 1/d_{\text{input}}$ | 1.0 |

Table 1: The table presents parameterizations of dense and MoE Transformers, showing parameter scaling in big-Θ notation, $d_{\text{input}}$ is the dimensionality of the weight input. Dense transformer $\mu$P is indicated in blue. MoE parameterizations build on dense $\mu$P. *simple*P MoE is heuristics marked in red, while the full-fledged theoretically grounded $\mu$P MoE is shown in green.

## 2 BACKGROUND AND RELATED WORK

**Mixture of Experts.** Mixture of Experts was originally introduced by (Jacobs et al., 1991), and later proposed in the context of language modeling by (Shazeer et al., 2017). This approach has since been successfully integrated into the Transformer (Vaswani et al., 2017) architecture in multiple works, including (Fedus et al., 2022; Lepikhin et al., 2020; Du et al., 2022; Zhou et al., 2022). In a Transformer model, the MoE layer is typically constructed by replacing the Feed-Forward component with a set of *experts*. Each expert retains the design of the original Feed-Forward layer, consisting of two linear layers with a non-linearity between them. Crucially, for any given input token, only a fraction of these experts are activated. The selection of experts for each token is determined by a routing mechanism - a simple linear layer followed by a softmax normalization and a Top-$k$ choice.

In the standard Switch layer (Fedus et al., 2022), each of the experts is of the same size as in the corresponding dense Transformer. This assumption is relaxed in fine-grained MoE (Dai et al., 2024; Ludziejewski et al., 2024), where for granularity $G$, the hidden size of each expert is reduced by a factor of $G$, while the number of experts and the router's top-$k$ value are both multiplied by $G$. In this way, the model has greater flexibility in mapping tokens to experts, while the total number and activated parameters remain approximately constant.

**Zero-shot hyperparameter transfer.** Standard Parametrization (SP) initializes weights with the usual variance scaling (like Xavier or Kaiming style: variance $\propto 1/d_{\text{in}}$ for many layers). SP learning rates do not scale with width, which means that as the model width increases, gradient magnitudes and update sizes can change unpredictably. Thus, SP often fails to preserve stability and hyperparameter transfer in scaling neural networks. To overcome this limitation, (Yang, 2021) introduced Maximal Update Parameterization ($\mu$-Parameterization, or $\mu$P). $\mu$P ensures that each layer in a network receives updates of the same order of magnitude during training, regardless of width. This allows for what is known as the feature learning regime, where internal representations evolve in a meaningful way as training progresses. Crucially, $\mu$P enables hyperparameter transfer across model sizes: one can tune learning rates and initialization on a small model and zero-shot transfer them to a large one, as shown empirically and theoretically in Yang et al. (2022). This paradigm, called $\mu$Transfer, has been shown to dramatically reduce the cost of training large models while maintaining performance. Later works (Yang et al., 2024; Everett et al., 2024) reformulate and generalize $\mu$P theory, while Dey et al. (2025) and Yang et al. (2023) include transfer across model depths. Despite its success on many architectures such as Transformers and ResNets, extending $\mu$P to Mixture-of-Experts models remains an open challenge. In this work, we address this problem.

## 3 PRINCIPLED APPROACH FOR $\mu$P FOR MOE

In this section, we analyze the training dynamics of Mixture-of-Experts (MoE) (Fedus et al., 2022) and derive a parameterization that ensures feature learning across all weights in the infinite width limit.

## 3.1 DEFINITIONS AND NOTATION

We scale the model width, with the number of experts and top-$k$ kept fixed. The hidden dimension of each expert grows proportionally with the model width, so that the ratio between them remains constant. We model the MoE layer as a Switch Transformer (Fedus et al., 2022), which consists of:

- A router matrix $R_0 \in \mathbb{R}^{n_{\text{experts}} \times n}$,
- Two weights of an MLP expert $E$: $E_1 \in \mathbb{R}^{n_{\text{experts}} \times 4n \times n}$, $\quad E_2 \in \mathbb{R}^{n_{\text{experts}} \times n \times 4n}$.

Here $n$ denotes the model width, which gets scaled to infinity.

The forward pass of the MoE layer is computed as:

$$\text{MoE}(x) = E(x)^T R(x).$$
$$E(x) = E_2 \text{ReLU}(E_1 x),$$
$$R(x) = \text{top-k}(\text{softmax}(R_0 x)).$$

We adopt the notation of Yang et al. (2022). A vector $v \in \mathbb{R}^n$ is said to be $\Theta(n^a)$ if

$$\frac{\|v\|^2}{n} = \Theta(n^{2a}),$$

where $\| \cdot \|$ is the Euclidean norm. Intuitively, this means that a typical entry of $v$ has magnitude $\Theta(n^a)$. Analogous definitions apply for $O(n^a)$ and $\Omega(n^a)$, and extend naturally to matrices.

All layers outside of MoE blocks are assumed to follow the TP5 parameterization.

## 3.2 INTUITION

TP5 categorizes weight matrices into three types based on their input and output dimensions. Sufficient conditions are:

- **Input weight:** maps fixed-size $\rightarrow$ unbounded; initialized as $\Theta(1)$; gradient updates of size $\Theta(1)$.
- **Hidden weight:** maps unbounded $\rightarrow$ unbounded; initialized as $\Theta(1)$; gradient updates of size $\Theta(1/n)$.
- **Output weight:** maps unbounded $\rightarrow$ fixed-size; initialized as $\Theta(1)$; gradient updates of size $\Theta(1)$.

By inspection, the expert weights $(E_1, E_2)$ map unbounded to unbounded dimensions, suggesting they should behave as *hidden weights*. The router weight $R_0$ maps from unbounded to fixed-size, making it a candidate *output weight*. We verify this classification in Section 3.4 by analyzing initialization and gradient scaling.

## 3.3 DESIDERATA

Following TP5, a $\mu$-parameterized model must satisfy:

1. At initialization, all hidden representations $h(x)$ in the network should scale as $\Theta(1)$.
2. The model output logits $f(x)$ should be $O(1)$ at initialization.
3. After one optimization step, the changes to hidden representations $\Delta h(x)$ and output logits $\Delta f(x)$ should be $\Theta(1)$.

These conditions guarantee stable layer weights and outputs across model widths, ensuring feature learning (Yang et al., 2022).

## 3.4 DERIVATION

In this section, we use the parameterization from Table 2. It is analogous to the parameterization from Table 1, on which we performed our experiments (See Yang et al. (2022) Appendix B).

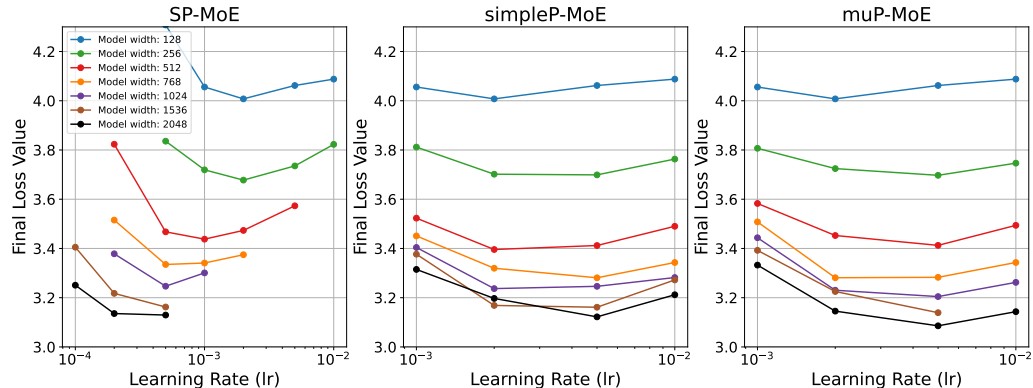

Figure 1: Learning rate transfer in MoE. **Left:** Standard Parameterization (SP). **Middle:** Our *simple*P, treating each expert as a feed-forward layer. **Right:** Our $\mu$P for MoE with both router and expert reparametrization. Under SP, the optimal learning rate depends strongly on width, while both reparameterizations achieve transfer across widths.

| Weight type | Init. Var. | LR (Adam) |
|---|---|---|
| Input weight | $1/d_{\text{input}}$ | $1.0$ |
| Output weight | $1/d_{\text{input}}^2$ | $1/d_{\text{input}}$ |
| Hidden weight | $1/d_{\text{input}}$ | $1/d_{\text{input}}$ |

Table 2: Parameterization from TP5 used in our derivation. Changes introduced by $\mu$P are highlighted in blue; quantities unchanged from SP are shown in gray.

### 3.4.1 INITIALIZATION

If we initialize $E_1, E_2$ as in a dense MLP, and set the router using the same variance and scaling factor as the output layer in TP5, we obtain:

If $E_1, E_2$ are initialized like *hidden weights*, and $R_0$ with the variance of an *output weight* (See Table 2), thus we obtain:

- $E(x) = \Theta(1)$
- $R_0 x = O(1)$
- $R(x) = \Theta(1)$
- $\text{MoE}(x) = \Theta(1)$

The term $R_0 x$ is non-standard, but it does not affect the overall output scale, which remains $\Theta(1)$.

### 3.4.2 OPTIMIZER STEP

Assuming all other modules satisfy the desiderata in Section 3.3, gradients with respect to hidden-layer outputs are $\Theta(1/n)$. For MoE, this implies:

$$\nabla\text{MoE}(x) = \Theta(1/n).$$

Now we can calculate $\nabla E(x)$:

$$\begin{aligned}
\nabla E(x) &= R(x)\nabla\text{MoE}(x)^T \\
&= \Theta(1) \cdot \Theta(1/n) = \Theta(1/n)
\end{aligned} \tag{1}$$

Here, the summation is over a finite number of experts, so it does not change the asymptotic scaling, i.e., the $\Theta(\cdot)$ order remains the same. Since $R(x)$ and $\text{MoE}(x)$ are 1-dimensional vectors, each entry of $\nabla E(x)$ is simply the product of the corresponding entry sizes.

Similarly calculate $\nabla R(x)$:

$$\begin{aligned}
\nabla R(x) &= E(x)\nabla\text{MoE}(x) \\
&= \Theta(1) \cdot \Theta(1/n) \cdot n = \Theta(1)
\end{aligned} \tag{2}$$

The multiplication by $n$ in this equality follows from $E(x)$ and $\nabla\text{MoE}(x)$ being correlated. We prove that correlation in Appendix B. In short, for correlated vectors $v, u \in \mathbb{R}^n$ quantity $v^T u$ has expected size $\Theta(v)\Theta(u) \cdot \text{corr}(v, u) \cdot n$, which follows from the Law of Large Numbers.

$\nabla E_1 x, \nabla E_2 \text{ReLU}(E_1 x)$ are $\Theta(1/n)$ since they mimic standard MLP layers. The Softmax and top-k over constant k do not change the size of the gradient For $R(x)$, since $R_0 x$ is $O(1)$, the gradient over $R_0$ is still $\Theta(1)$. That means $E_1, E_2, R_0$ receive the same gradient sizes as hidden weights and output weight, respectively, so they behave in the same way in training as their weight types from TP5

In conclusion, $E_1, E_2$ should be parameterized as *hidden weights*, while $R_0$ should be treated as an *output weight*.

## 4 EXPERIMENTAL RESULTS AND ALTERNATIVE VIEWS ON HYPERPARAMETER TRANSFER IN MOE

This section presents experimental results on learning rate transfer in MoE Transformers, providing empirical validation of our parameterization. Additionally, we explore the transferability of learning rate when scaling different MoE dimensions.

### 4.1 ALTERNATIVE PARAMETERIZATION FOR MOE MODELS

In Section 3, we develop a theory for $\mu$P for MoE where we re-parametrize both router and experts. In addition to this theoretically grounded scheme, we also evaluate a simplified parameterization, which we call *simple*-Parameterization (*simple*P). *simple*P applies the $\mu$P rules of dense Transformers directly to each expert, leveraging the structural similarity between experts and Transformer MLP blocks. The router, however, is left unmodified. Table 1 summarizes both parameterizations. Importantly, *simple*P is not a $\mu$-Parameterization, as it lacks theoretical guarantees for feature learning.

### 4.2 MODEL WIDTH

We evaluate the transferability of the learning rate across model widths (Figure 1 and Figure 4 in Appendix). For both $\mu$P and *simple*P, the optimal learning rate shifts slightly upward as width increases. A similar trend appears in Tensor Programs 5 and in our dense model experiments (Figure 3). We hypothesize that this shift arises from instabilities in architectures with large depth-to-width ratios, which may underperform on high learning rates. A more detailed analysis is left for future work. Overall, these results support the transferability of learning rates under both $\mu$P and *simple*P.

### 4.3 COMPUTATIONAL SAVINGS IN TUNING

To illustrate the efficiency gains from $\mu$Transfer, consider training a model with width 2048. A full run requires roughly $4 \times 10^{18}$ FLOPs. In contrast, tuning on a smaller model of width 128 requires only $1.5 \times 10^{16}$ FLOPs, over 250 times cheaper. Thus, tuning costs become negligible compared to the final training run, underscoring the practical value of $\mu$Transfer.

### 4.4 SCALING OTHER MOE DIMENSIONS

In the previous section, we showed MoE parameterizations for varying model widths. In this section, we investigate whether scaling the MoE architecture in two other dimensions necessitates reparameterization.

**Number of experts.** Increasing the number of experts expands model capacity without increasing computational cost (Clark et al., 2022; Ludziejewski et al., 2025). As shown in Figure 2(a), models with more experts achieve lower final loss, while the optimal learning rate shifts slightly downward. The shift is small and comparable to that observed under width scaling, though it may indicate a more fundamental effect. At the scale of our experiments, however, verifying a dedicated parameterization for expert scaling is infeasible, as the results under SP remain within the margin of error. Overall, we find that changing the number of experts does not alter the stability of the under $muP$ for MoE parameter transfer.

**Granularity.** Granularity, described in Section 2, adjusts expert size while keeping computational cost fixed. Increasing granularity within a reasonable limit improves the model performance (Ludziejewski et al., 2024). It can be used in practice to match hardware constraints (Dai et al., 2024). Figure 2(b) shows that granularity changes break learning rate transfer. We attribute this to changes to scaling top-$k$ or the reduced hidden dimension of each expert. Since both settings alter router output dimensions, they lie outside the assumptions of our theory. We leave a full theoretical treatment of these results for future work.

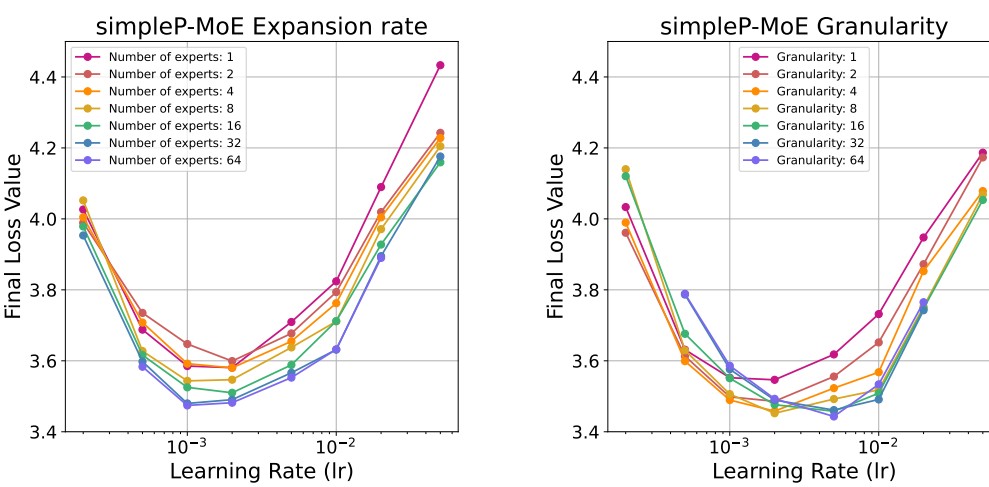

Figure 2: (a) Varying the number of experts. Given our $\mu$P for MoE, the optimal learning rate is preserved across a varied number of experts. (b) Varying granularity. Learning rate is not preserved across different granularities.

## 5 CONCLUSIONS

We introduced a $\mu$-Parameterization for Mixture-of-Experts (MoE) that preserves feature learning across widths by classifying expert weights as hidden and the router as output within the TP5 framework. Our theory implies stable gradient scales and width-invariant training dynamics, and our experiments confirm that the optimal learning rate transfers reliably across model sizes under our $\mu$P. Moreover, a simplified expert-only parameterization *simple*P also enables learning rate transfer in the scope of our experiments.

Beyond width, we examined other MoE scaling axes. Increasing the number of experts improves final loss with only a minor shift in the optimal learning rate, comparable to width scaling. In contrast, changing granularity breaks learning-rate transfer, likely due to altered router behavior and expert hidden sizes—conditions outside our current assumptions. These findings delineate where zero-shot hyperparameter transfer is robust in MoE, and where reparameterization or new theory is needed the most.

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

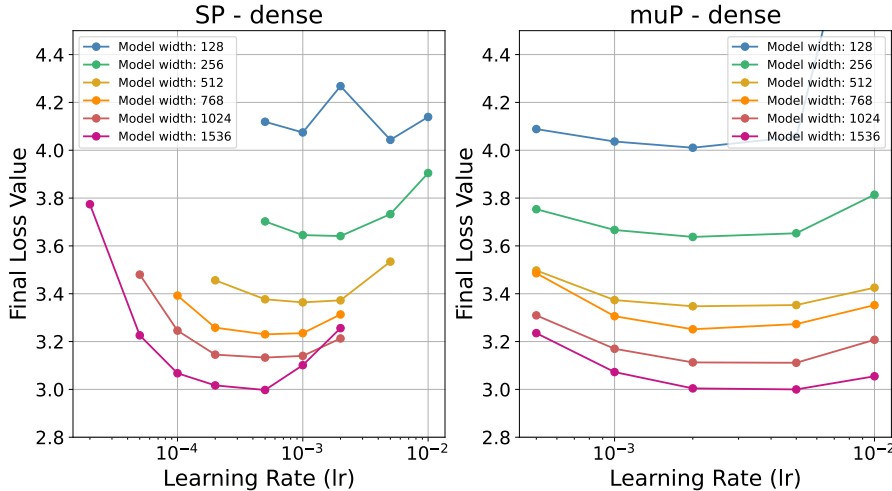

Figure 3: This figure shows experiments on learning rate transfer in *dense* models. SP on the left has a different optimal learning rate for each model width, while $\mu$P has a mostly stable optimum (with slight upward shift, same as TP5 and MoE sweeps).

## A   MuP FOR DENSE TRANSFORMER

In this section, we verify the findings from Yang et al. (2022) by implementing the $\mu$P for dense models. In contrast to standard parameterization, for $\mu$P, the optimal learning rate transfers between different model widths.

## B   EXPERT–GRADIENT COVARIANCE LEMMA

We now formalize the intuition that in a µP-parametrized Switch-MoE block, each active expert's forward activation correlates with its backward gradient at order $\Theta(1/n)$, and that the router's gradient norm remains $\Theta(1)$ both immediately after initialization and again after one µ-SGD/Adam update.

**Lemma B.1** (Expert–gradient covariance). *Let an L-block Switch-MoE be µP-parametrized with width $n \to \infty$, a fixed number of experts $n_{\text{experts}} = O(1)$, and fixed top-$k = O(1)$. For each block $\ell$, define*

$$y^{(\ell)} = \sum_{e=1}^{n_{\text{experts}}} R_e^{(\ell)}(x^{(\ell)}) E_e^{(\ell)}(x^{(\ell)}),$$

$$\delta^{(\ell)} = \nabla_{y^{(\ell)}} L \in \mathbb{R}^n. \tag{3}$$

*Assume the inductive hypothesis*

$$\frac{1}{n}\sum_{j=1}^n (\delta_j^{(\ell)})^2 = \Theta(n^{-2}) \implies \delta_j^{(\ell)} = \Theta(n^{-1}) \quad \text{for typical } j. \tag{H$_\ell$}$$

*Then for every block $\ell$ and any active expert $e$, at both*

$$t = 0 : \text{ immediately after initialization},$$
$$t = 1 : \text{ after one } \mu\text{-SGD/Adam step},$$

*we have*

$$\text{Cov}(E_{e,j}^{(\ell)}, \delta_j^{(\ell)}) = \Theta(n^{-1}),$$
$$\left\| \nabla_{r^{(\ell)}} L \right\|_2 = \Theta(1). \tag{4}$$

*Proof.* **Notation.** For block $\ell$, set

$$\mathbf{e}_e = E_{\ell,e}^{(2)} \operatorname{ReLU}\!\big(E_{\ell,e}^{(1)} x^{(\ell)}\big) \in \mathbb{R}^n,$$
$$R_e = R_e^{(\ell)}\big(x^{(\ell)}\big) \in \mathbb{R}, \tag{5}$$
$$\delta = \delta^{(\ell)} \in \mathbb{R}^n,$$
$$J^{(\ell)} = \frac{\partial y^{(\ell)}}{\partial x^{(\ell)}} \in \mathbb{R}^{n \times n},$$

so that $\delta^{(\ell-1)} = (J^{(\ell)})^\top \delta$ and $J_{ij}^{(\ell)} \sim \mathcal{N}(0, 1/n)$ under µP.

**Step 1: Stein's lemma (holds at $t=0$ and $t=1$).** Fix expert $e$, coordinate $j$, and define

$$Z = \mathbf{e}_{e,j} \sim \mathcal{N}(0, \sigma^2),$$
$$c = \sum_{e' \neq e} R_{e'} \, \mathbf{e}_{e',j}, \tag{6}$$
$$g(z) = [L'(y^{(\ell)})]_j,$$
$$y_j^{(\ell)} = R_e \, z + c.$$

Then $g'(z) = R_e \, [L''(y^{(\ell)})]_j$ and by equation $\mathrm{H}_\ell$, $[L''(y^{(\ell)})]_j = \Theta(n^{-1})$. Thus

$$\begin{aligned}
\operatorname{Cov}(Z, g(Z)) &= \sigma^2 \, \mathbb{E}\big[g'(Z)\big] \\
&= \sigma^2 \, R_e \, \mathbb{E}\big[L''(y^{(\ell)})\big]_j \\
&= \Theta(1) \cdot \Theta(1) \cdot \Theta(n^{-1}) = \Theta(n^{-1}).
\end{aligned} \tag{7}$$

**Remark.** Because by induction the block-$(\ell+1)$ Hessian entries already scale like $\Theta(n^{-1})$, and passing any such matrix back through a µP-linear layer (whose weights are $\mathcal{N}(0, 1/n)$) multiplies each term by another $1/n$ but sums over $n$ of them, the net effect is still $\Theta(n^{-1})$. In other words, a $1/n$ factor per weight-matrix multiplication exactly preserves the $\Theta(n^{-1})$ scale of $\big[L''(y^{(\ell)})\big]_j$.

**Step 2: Router-gradient norm (holds at $t=0$ and $t=1$).** Summing the covariances over $j$,

$$\mathbb{E}\big[\mathbf{e}_e^\top \delta\big] = \sum_{j=1}^n \operatorname{Cov}(\mathbf{e}_{e,j}, \, \delta_j) = n \cdot \Theta(n^{-1}) = \Theta(1). \tag{8}$$

Since $\operatorname{Var}(\mathbf{e}_e^\top \delta) = O(n^{-1})$, Chebyshev's inequality gives $\mathbf{e}_e^\top \delta = \Theta(1)$ with high probability, i.e. the second line of equation 4.

**Step 3: One-step update.** Under µ-SGD/Adam with LR $\eta/n$ on experts and $\eta$ on router, the factors in equation 7 and equation 8 change by at most a $(1 + O(n^{-1}))$ factor, so both lines of equation 4 hold at $t=1$.

**Step 4: Depth induction.** Using $\delta^{(\ell-1)} = (J^{(\ell)})^\top \delta^{(\ell)}$ and $J_{ij}^{(\ell)} \sim \mathcal{N}(0, 1/n)$, one shows $\frac{1}{n} \sum_j (\delta_j^{(\ell-1)})^2 = \Theta(n^{-2})$, establishing $(H_{\ell-1})$. The base case $\ell = L$ is given by Tensor-Programs V; induction completes the proof. $\square$

## C  EXPERIMENTAL SETUP

All models in this study are decoder-only Transformers trained on the C4 dataset (Raffel et al., 2020). We use the GPT-2 tokenizer (Radford et al., 2018) and optimize with AdamW (Loshchilov & Hutter, 2019). Training follows a cosine decay schedule with linear warmup for the first $1\%$ of steps. Weights are initialized with a normal distribution, as the theory of Tensor Programs assumes (Yang et al., 2022). Mixed precision training is applied, with Attention component computed at high precision. The models employ MLP with ReLU activations. MoE models are Switch Transformers (Fedus et al.,

2022). As a standard MoE setup we used $8$ Experts, $1$ of which is activated per token. All models have Attention head dimension of $64$. Two auxiliary losses are used for the Router: a z-loss weighted at $0.001$ (Zoph et al., 2022) and load balancing weighted at $0.01$ (Fedus et al., 2022).

Models for Figure 1 are trained for 5.2B tokens with 16 Transformer layers ($16\times$ attention + feed-forward). Dense models for Figure 3 have $24$ layers and are trained for 16B tokens. Models for Figure 2 have 12 blocks and are trained for 2.5B tokens. Both dense and MoE width sweeps use a base width of $128$. In Figure 1, the runs with width $128$ are reused across all three panels since, when model width equals the base width, all parameterizations coincide. Additionally, In Fig. 4 the models have 8 transformer layers and are trained for 1B tokens.

Differences in total tokens and layer counts across figures arise from experiments conducted at different times under changing compute budgets. MoE runs are smaller/shorter to enable one additional width; we do not expect these scale differences to affect the qualitative conclusions.

The expert-count and granularity ablations are run under *simple*P at base width $256$ and model width $768$. Because width is fixed within each sweep, the parameterization only induces a global shift in the learning-rate scale and does not affect relative comparisons; the same conclusions would hold under SP or $\mu$P.

## D    MORE RESULTS

Fig. 4 presents the smaller models with 8 transformer layers and are trained for 1B tokens. The results remain consistent with the findings from the main paper. The perfromance transfers under both *simple*P and $\mu$P parameterizations.

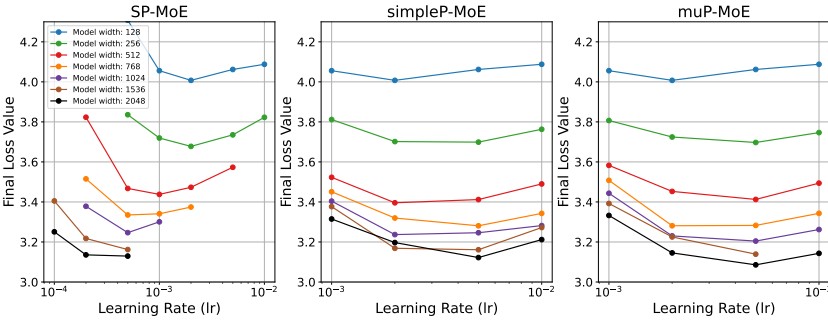

Figure 4: The plots present MoE performance for varying learning rates in the following set-ups: standard parametrization (SP) with no scaling on the left. *simple*P - treating each Expert like a FeedForward layer in the middle. $\mu$P - our theory applied to MoE layer on the right. While in the case of SP, the optimal learning rate is different for different model sizes, both reparameterizations achieve learning rate transfer across model widths.

