# OpenReview forum: "$\mu$-Parameterization for Mixture of Experts"
_ICLR.cc/2026/Conference — ICLR 2026 Conference Withdrawn Submission_

### Official Review · Reviewer_Qdp6 · 2025-10-20

**Soundness:** 2
**Presentation:** 3
**Contribution:** 1
**Rating:** 2
**Confidence:** 3

**Summary:**

The authors propose an extension of $\mu$-parameterization for MoE architectures, enabling transfer of optimal learning rates across increasing MoE model width. The authors provide a theoretically grounded derivation of their parameterization, along with limits for which the theoretical guarantees cease to hold, such as for varying MoE granularity.

**Strengths:**

The paper discusses an important practical and empirical topic -- hyperparameter transfer across model scale -- and uses principled foundations to derive theoretical results with accompanying guarantees and limitations. The paper is also clear and the logic is easy to follow.

**Weaknesses:**

The main weakness of the paper is the extremely limited experimental validation. The authors only present experiments for one single hyperparameter, the learning rate, in one single MoE model, Switch transformer, for one dataset, C4. The theoretical work is interesting but with such limited experimental results it becomes infeasible to really assess the impact and empirical consequences of the work. On a related note, the paper is short at only 6 pages, so I'm curious as to why the authors didn't consider using some of the ample additional page limit to more extensively verify their work.

The technical novelty is also limited, it being an extension of $\mu$-parameterization [Yang et al, NeurIPS 2021] to MoE architectures. In my view, extending theoretical works or methods centered on dense models to MoE is still worthwhile and interesting, but it does then require a thorough analysis of how MoE presents new challenges and opportunities. In this work, however, the analysis appears substantially more limited than in Yang.

**Questions:**

Is there a reason for the paper being so short, and the experimental validation being so limited? There are numerous datasets, modalities, and models you could have considered, for which experimental results would help support your work.

---

### Official Review · Reviewer_FdRu · 2025-10-26

**Soundness:** 3
**Presentation:** 3
**Contribution:** 3
**Rating:** 6
**Confidence:** 3

**Summary:**

This paper addresses the challenge of hyperparameter tuning in extremely large Language Models (LLMs), particularly those utilizing the Mixture-of-Experts (MoE) architecture. MoE has emerged as a leading architecture for scaling LLMs, but existing $\mu$-Parameterization ($\mu$P) techniques, which enable transfer of optimal hyperparameters across model scales ($\mu$Transfer), have previously focused only on dense architectures.
The authors derive a theoretically grounded $\mu$-Parameterization for MoE, building upon the Tensor Programs 5 (TP5) framework. The core theoretical finding classifies the expert weights ($E_1, E_2$) as hidden weights and the router weight ($R_0$) as an output weight within the $\mu$P framework. The paper empirically verifies that this $\mu$P scheme successfully transfers the optimal learning rate across varying model widths (up to 2048). Furthermore, the authors introduce a simplified parameterization (simpleP) that also achieves transferability, and they investigate other scaling axes, finding that while increasing the number of experts preserves transfer, changing MoE granularity breaks it.

**Strengths:**

**Originality**: The primary strength is the novel theoretical derivation and empirical validation of $\mu$P for Mixture-of-Experts. This is a crucial extension of existing Tensor Program theory that had previously overlooked sparse architectures. The paper also introduces and evaluates simpleP, a heuristic parameterization, providing a useful comparison point.

**Quality**: The theoretical analysis is high quality, providing a principled method for scaling MoE components by classifying expert weights as hidden and the router as output, thereby ensuring stable gradient and activation scales across widths. The theoretical argument is supported by technical proofs regarding covariance (Appendix B). The empirical results, showing successful learning rate transfer for both $\mu$P and simpleP across a wide range of widths, are compelling.

**Clarity**: The paper is well-written, making complex theoretical concepts accessible. The tables summarizing the parameterization rules are particularly helpful.

**Significance**: This work directly addresses the compute bottleneck inherent in training LLMs over 1T parameters by enabling cost-efficient hyperparameter tuning. The ability to transfer optimal learning rates is a foundational step toward efficient large-scale MoE training.

**Weaknesses:**

**Missing critical information for experiments**: It’s unclear from the text what is the range of model sizes considered in the experiments. Can the authors add a table on the sizes of models considered in terms of overall number of parameters and number of active parameters?

**Questions:**

**Utility of $\mu$P over simpleP**: Since simpleP (heuristic, experts only reparameterized) empirically shows strong learning rate transferability similar to the full $\mu$P (router and experts reparameterized), what practical benefits does the theoretically grounded $\mu$P provide over simpleP? Do the theoretical guarantees translate into demonstrably better final model performance, faster convergence, or improved stability during training runs, especially at the largest tested widths?

**Clarification on Figures**: The authors in lines 110-112 claim “We scale the model width, with the number of experts and top-k kept fixed. The hidden dimension of each expert grows proportionally with the model width, so that the ratio between them remains constant.” Does this mean that for results in Figure 1 and Figure 2a, both the model dimension and number of experts are being scaled simultaneously? Or in Figure 1, only model dimension is increased with fixed number of experts and in Figure 2, only number of experts is increased with model dimension fixed?

---

### Official Review · Reviewer_HLwj · 2025-10-30

**Soundness:** 1
**Presentation:** 1
**Contribution:** 1
**Rating:** 2
**Confidence:** 3

**Summary:**

This addresses the substantial cost associated with hyperparameter tuning in extremely large-scale Language Models (LLMs), particularly those utilizing the Mixture-of-Experts (MoE) architecture.

**Strengths:**

1. The paper solves a major engineering bottleneck for large-scale AI research. By enabling the transfer of optimal learning rates across MoE model scales.

**Weaknesses:**

1. The paper is not well-organized and presented. It would be better to write the texts and give more beautiful pictures for this venue.
2. The simpleP parameterization is shown to be highly effective at learning rate transfer, performing similarly to $\mu$P. A more in-depth discussion is needed to theoretically justify why simpleP, which is less complex than the full $\mu$P derivation, works so well, and what specific scenarios would necessitate the complexity of the full $\mu$P.

**Questions:**

1. Could the authors provide the results for the expert-count and granularity ablations (currently run under simpleP) using the proposed $\mu$-Parameterization ($\mu$P)?
2. The simpleP parameterization seems to achieve learning rate transfer very successfully16. Could the authors provide a brief theoretical analysis or justification for why simpleP works effectively in this context, and explain the key non-trivial theoretical differences that make $\mu$P the preferred choice over simpleP for massive-scale MoE training?
3. While the paper focuses on the learning rate, $\mu$-Transfer is typically used to transfer other critical hyperparameters (e.g., initialization scale, weight decay). Do the theoretical guarantees of the derived $\mu$P extend to these other hyperparameters in MoE architectures? If so, could the authors provide preliminary empirical evidence demonstrating the successful transfer of one additional hyperparameter (e.g., weight decay) across model widths using $\mu$P?

---

### Note · Authors · 2025-12-03

**Comment:**

\section*{Combined Response}

We thank all reviewers for their time and thoughtful comments. We appreciate the feedback and agree with some of the weaknesses raised. Below we address the main points raised across the reviews. We also state our decision to withdraw the submission.

\paragraph{1. simpleP vs.\ $\mu$P in MoE (Reviewers R1 \& R2).}
Reviewers noted that simpleP performs nearly as well as $\mu$P in our experiments, raising the question of what practical benefits the theoretically grounded $\mu$P provides.

To clarify: the only structural difference is that $\mu$P reparameterizes the router matrix as an output weight, whereas simpleP leaves the router unscaled. According to TP5 theory, leaving the router unscaled leads to diverging updates in the infinite-width limit, which is why simpleP is theoretically ungrounded.

However, at the moderate widths we were able to explore (up to model dimension~2048), this instability does not manifest. The gap between $\mu$P and simpleP is negligible at our scale, and we cannot reliably speculate at what scale the divergence would become observable.

\paragraph{2. Experimental details (R1 \& R2).}
Some reviewers asked why the expert-count and granularity ablations were run only under simpleP.

The reason is that these ablations do not modify model width, and therefore parameterization has no effect on the dynamics. In these sweeps, simpleP and standard parameterization behave identically, and $\mu$P does not apply because its derivation assumes a fixed number of experts (assumption broken in both ablations).

We acknowledge that parts of the experimental setup were insufficiently explained in Appendix~D, and we agree that a clearer presentation with model details tables would improve the paper.

\paragraph{3. Limited experimental scope (R1 \& R3).}
We agree that our empirical exploration covers only a limited fraction of the full search space. Given our compute constraints, we made a conscious tradeoff: we prioritized training larger models for longer, rather than spreading the same budget across additional datasets, modalities, or many small-scale ablations. We believe that longer, higher-capacity training runs provide more informative insights than duplicating experiments on additional datasets.

We also emphasize that we did run two full-width sweeps on MoE models: a smaller one included in the appendix (where the current version contains an incorrectly pasted plot), and a larger one (Figure~1) with 16 layers trained for 5.2B tokens, reaching approximately 0.8B active and 4.5B total parameters at the largest width. That said, we agree that adding additional experimental angles—such as transferring a second hyperparameter—would strengthen the work.

\paragraph{4. Paper length (R3).}
We acknowledge R3’s point regarding the short length of the submission (6 pages). We opted for a concise presentation, but we agree that this brevity made the contribution appear weaker.

\section*{Decision}
We respectfully withdraw the submission.

We appreciate the reviewers’ constructive comments and plan to expand the experimental section, present a clearer theoretical comparison between $\mu$P and simpleP, and improve the structure and clarity in a future version of the work.

Thank you again for your time.

**Withdrawal Confirmation:**

I have read and agree with the venue's withdrawal policy on behalf of myself and my co-authors.